# Comparison between Chemical and Biological Degradation Processes for Perfluorooctanoic Acid

Xuhan Shu [1], Rama Pulicharla [1], Pratik Kumar [2] and Satinder Kaur Brar [1,*]

1 Department of Civil Engineering, Lassonde School of Engineering, York University, North York, Toronto, ON M3J 1P3, Canada; shineshu@yorku.ca (X.S.); ramapuli@yorku.ca (R.P.)
2 Department of Civil Engineering, Indian Institute of Technology Jammu, Jagti, NH 44, Nagrota Bypass, Jammu 181221, India; pratik.kumar@iitjammu.ac.in
* Correspondence: satinder.brar@lassonde.yorku.ca; Tel.: +1-416-7362100-44703

**Abstract:** Perfluorooctanoic acid (PFOA) is a perfluoro compound that contains an eight-carbon perfluoroalkyl chain followed by a carboxylic acid function group. The C-F bound possesses a strong bond energy of approximately 485 kJ/mol, rendering PFOA thermally and chemically stable. It has found applications in water-resistant coating and is produced either by degrading other long-chain perfluorinated carboxylic acids or fluorotelomer alcohol. PFOA is challenging to further degrade during water treatment processes, leading to its accumulation in natural systems and causing contamination. Research has been conducted to develop several methods for its removal from the water system, but only a few of these methods effectively degrade PFOA. This review compares the most common chemical degradation methods such as photochemical, electrochemical, and sonochemical methods, to the cutting-edge biodegradation method. The chemical degradation and biodegradation methods both involve the stepwise degradation of PFOA, with the latter capable of occurring both aerobically and anaerobically. However, the degradation efficiency of the biological process is lower when compared to the chemical process, and further research is needed to explore the biological degradation aspect.

**Keywords:** perfluorooctanoic acid (PFOA); perfluorochemicals (PFCs); biodegradation; chemical degradation; degradation products; defluorination



## 1. Introduction

### 1.1. Background and Characterization of the Perfluoro Compounds

Perfluoro chemicals (PFCs) are man-made chemical compounds that have seen extensive use in various industrial products. They are commonly found in cleaning agents, firefighting foams, and the non-stick coatings on cooking pans [1]. Due to the high bonding energy of the C-F bond, which is approximately 485 kJ/mol in perfluoroalkyl moieties, perfluoro chemicals exhibit strong chemical and thermal stability. This characteristic renders them persistent compounds within the natural environment [2]. When an alkyl chain exceeds eight carbons, it is classified as a long-chain PFC, whereas if it contains fewer than eight carbons, it falls under the category of short-chain PFC. During wastewater treatment processes, long-chain PFCs degrade into shorter-chain products [3]. Other perfluoro compounds, such as fluorotelomer alcohol (FTOH), polyfluoroalkyl phosphates (PAPs), 8:2 fluorotelomer acrylate (8:2 FTAC), and fluorotelomer carboxylates (FTCAs), can also degrade into perfluorooctanoic acid (PFOA) [4]. As chain length decreases, so does toxicity. However, because short-chain PFCs are more hydrophilic than long-chain PFCs, they are more prevalent in natural water systems [5]. In natural water systems, approximately 88.8% of PFC contamination comprises short-chain PFCs [2].

A primary category of compounds in PFCs is referred to as per- and polyfluoroalkyl (PFAS) compounds, as indicated by USEPA. These compounds consist of a variable-length fluorinated alkyl chain followed by a functional group, as depicted in Figure 1. Figure 1

illustrates that there are primarily two types of PFAS: perfluorinated sulfonic acids (PFSAs) and perfluorinated carboxylic acids (PFCAs). PFSA consists of a variable-length fluorinated alkyl chain followed by a sulfonic acid functional group. Upon degradation and defluorination, the long alkyl chain breaks down into short-chain perfluorooctanesulfonic acid (PFOS), which contains eight carbon atoms in the alkyl chain. Another type of PFAS is PFCA, which is formed by various alkyl chains connected to a carboxylic functional group. When long-chain PFCA undergoes degradation, the length of the alkyl chain decreases, and one of the principal degradation products is PFOA [1].

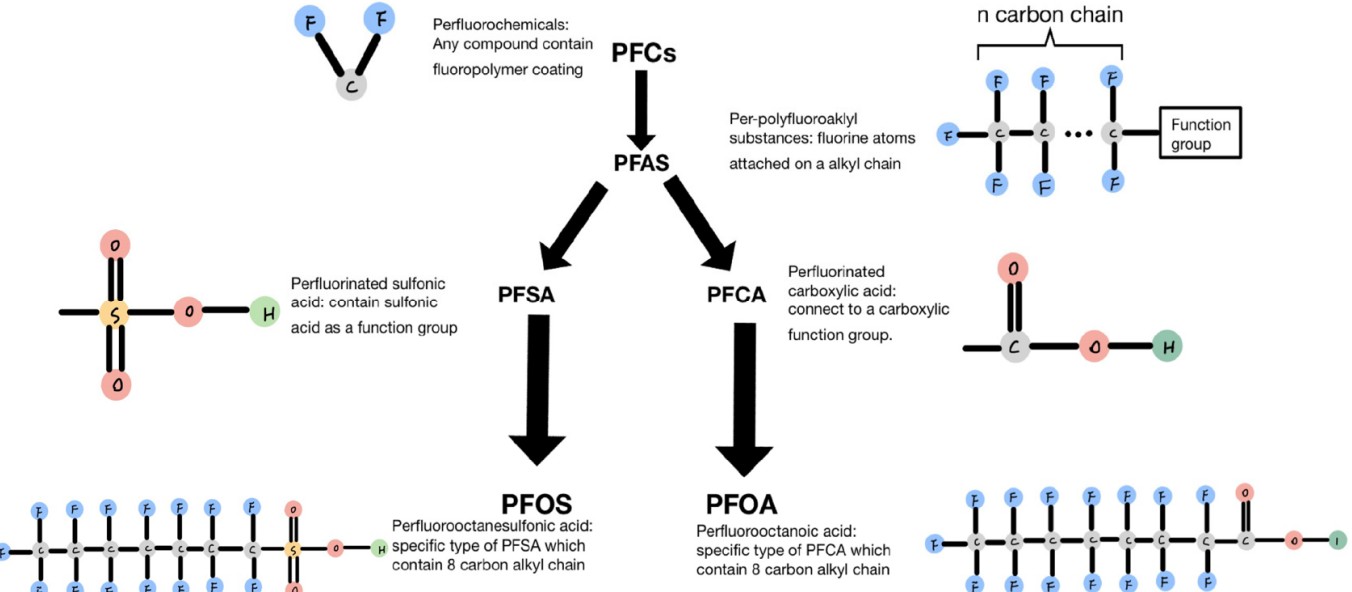

**Figure 1.** Classification and characteristics of each compound.

Both PFOA and PFOS are common perfluoro compound contaminants that have been widely detected in the environment. This article will specifically focus on PFOA, which exists as a white waxy powder at room temperature. PFOA is utilized in the production of Teflon for non-stick cookware and is also employed in the manufacturing of surfactants such as shampoo and floor wax, taking the form of fluoropolymers and telomer alcohols. Following their use and disposal, these chemical compounds can eventually degrade into PFOA [6]. Due to its solubility in water (9.5 g/L) and its extended half-life in water systems (92 years), PFOA rarely degrades in the natural environment and is often referred to as a 'forever chemical' [7]. After PFOA is released from industrial and wastewater treatment plants, some of it is emitted into the atmosphere, while the rest travels through surface-water and groundwater systems [8]. It accumulates within the biota through the food chain and ultimately enters the human body through the consumption of food or drinking water [9]. The research indicates that the aquatic toxicity of PFOA for marine invertebrates is approximately 10–24 ppm, which is several orders of magnitude higher than the detected PFOA concentration in natural systems [10]. However, it has the potential to be transported through the food chain and ultimately bioaccumulate in aquatic life and humans. Recent studies have shown that trace concentrations of PFOA have been detected in human serum worldwide [9]. Typically, it targets sensitive organs such as the liver or kidney. When PFOA combines with albumin, it can reduce glomerular filtration in the kidney or lead to liver hypertrophy or necrosis [11]. The acute toxicity of PFOA in humans is still unknown, but the lethal dose (LD50) in male rats is 175–208 mg/kg [12]. The reference dose (RfD) for PFOA is 20 ng/kg/day, and an overdose can lead to accumulation in the human body, potentially increasing the risk of cancer [11]. To monitor and regulate this emerging contaminant, the US Environmental Protection Agency (USEPA) proposes healthy drinking water advisory levels for PFOA at 70 ng/L. In 2018, Health Canada published a drinking

water standard of 200 ug/L for PFOA [13]. PFOA can cause aquatic toxicity, affecting the survival, growth, and reproduction of aquatic flora and fauna, and it can also be harmful to human health [10]. Due to PFOA's toxicity, there is increasing attention on it as an emerging contaminant. Various methods have been developed to remove these contaminants and control pollutant concentrations to ensure they remain within safe levels.

*1.2. Physical Removal Method and Degradation Methods*

Physical removal methods, such as granular activated carbon adsorption, ion exchange resin, membrane filtration, and coagulation, have been frequently documented [1,14–16]. Recent research has demonstrated that ion exchange resins exhibit high adsorption capacities (525–1500 mg/g) in comparison to granular activated carbon adsorption (41–120 mg/g), and both methods can achieve over 90% removal [17]. These physical removal techniques have been developed at a full scale, making them applicable in drinking water treatment plants for PFOA removal. However, their primary drawback is that they solely extract contaminants from water resources and concentrate them without degradation [18]. Therefore, an additional degradation method or step is necessary to eliminate these compounds and prevent the introduction of secondary contaminants.

There are three primary methods for degrading these compounds. The first is a nonchemical degradation method, such as plasma technology, incineration, sonolysis, and photolysis. These methods require high-energy inputs, such as UV, ultrasound, or heat, to initiate degradation. During the reaction, the input energy can break down the C-F bond and convert PFOA into $CO_2$, free fluoride ions, and less toxic short-chain PFCA. This reaction typically occurs in a high-temperature, high-pressure environment without the involvement of any chemical compounds [19]. The second method is chemical degradation, which involves chemicals like ferric ions or persulfate as catalysts in the reaction. These chemicals lower the activation energy needed for the bond-breaking process, accelerating the degradation reaction [20]. The third degradation method is biological degradation, which represents a newer technology. During biodegradation, microorganisms consume PFOA as a carbon source [21]. Biodegradation boasts lower energy consumption and does not yield unpredictable by-products during the reaction [22].

*1.3. Research Gap and Innovation*

The chemical degradation method for PFOA has been well developed on a laboratory scale. There are numerous review papers and reports available for comparing different types of chemical degradation [20,23]. However, when it comes to the biodegradation of PFOA, which represents a frontier technology, there are only a few review articles available [24,25]. To date, there has been no article comparing the biodegradation and chemical degradation methods of PFOA. To address this research gap, the proposed review will discuss the comparison between chemical and biological degradation for PFOA, examining aspects such as the reaction environment, mechanisms, degradation by-products, and efficiency of different degradation methods. This review article will enable readers to gain a better understanding of various chemical degradation methods and cutting-edge biological degradation methods. It will also assist researchers in comprehending the advantages and disadvantages of these methods and exploring potential avenues for future research.

## 2. Chemical Degradation

Chemical degradation is a well-established technology widely utilized in wastewater treatment systems. In most cases, it requires an energy source input, such as UV, electrical power, ultrasound, and microwave energy. With the presence of chemical catalysts, it generates free electrons or free radicals to break down the C-F bond in PFOA for degradation. The most employed chemical degradation methods include photochemical and electrochemical degradation [26,27]. Other forms of energy input are also used for degradation, including sonochemical degradation, microwave-enhanced degradation, and other chemically catalyzed degradation methods [28–30].

## 2.1. Photochemical Degradation

Most of the research on chemical degradation involves using UV as the input energy source to facilitate homogeneous or heterogeneous reactions [31–33]. Table 1 shows a selection of photocatalytic degradation studies that focused on low-pressure-to-medium-pressure mercury UV lamps with a wavelength of around 256 nm. This is because PFOA exhibits maximum absorbance at this wavelength, providing optimized energy for breaking down the chemical bonds in PFOA during the degradation process [33,34]. The UV lamp power varies from 9 W to 200 W, but generally, it utilizes low-power light, typically around 20 W, to maximize energy efficiency. [35] used a 15 W UV lamp with a wavelength of 240 nm to perform an anaerobic reduction reaction. Without the presence of chemicals as a reducing agent, the PFOA concentration only reduced by 13.3% in 6 h in an anaerobic environment. However, with the addition of KI as a catalyst, the PFOA concentration dropped to 93.9% in 6 h [35].

Table 1 demonstrates that photochemical degradation typically occurs at room temperature under a room atmosphere. The aerobic reaction takes place in a lower pH range, typically from 0.3 to 4.5, while the anaerobic reaction occurs in a higher pH range, usually between 8 and 12 [33,35]. There are two primary reasons for maintaining the aerobic reaction in an acidic environment. Firstly, the aerobic environment employs oxidation reagents to generate hydroxyl radicals for degradation. However, when the pH exceeds 4, it can render the oxidation agent unstable. For instance, at high pH levels, $H_2O_2$ degrades into $H_2O$ and $O_2$, reducing the formation of free radicals. The second reason is that the precipitation of $Fe(OH)_3$ forms at high pH, which can obstruct ongoing reactions and significantly impede the degradation performance. Therefore, the reaction should occur in a low-pH environment [36]. An experiment conducted by Song et al. (2013) involved reductive defluorination of PFOA under anaerobic conditions. The results indicated that increasing the pH from 8.1 to 10.3 enhanced the defluorination from 58% to 85%. This is because free electrons drive a reduction reaction in the experiment, and the relative quasi-stationary concentration of eaq- (ROSC) is linked to pH. With an increase in pH, more free electrons are available in the solution for defluorination to occur [37].

The photocatalyst plays a crucial role in the photochemical degradation of PFOA. Wu et al. (2017) used ZnO as a catalyst that absorbs PFOA for degradation. However, an increase in pH reduces the adsorption of PFOA on the reaction surface and consequently decreases the reaction rate [38]. Furthermore, during application, elevated temperatures reduce the surface area of ZnO, leading to a reduction in reaction efficiency [38]. $TiO_2$ is another popular catalyst employed in photochemical degradation. It contributes electrons during the reduction reaction and provides a reaction surface [39]. The effectiveness of the TiO2 surface is influenced by pH, leading to changes in degradation efficiency [39]. Araniti et al. (2015) utilized Mg-aminoclay-coated nanoscale zero-valent iron as a catalyst for PFC degradation. After aging for 3 days, the degradation rate decreased by 15% due to the reduced amount of reactive iron in the coating [40].

The main degradation products of PFOA through photochemical degradation are fluoride ions, $CO_2$, and other short-chain PFOA compounds [41]. Perfluoroheptanoic acid (PFHpA) is the initial degradation by-product that emerges during the photochemical degradation of PFOA [42]. Throughout the degradation process, PFOA undergoes stepwise defluorination and shortens its alkyl chain, resulting in the production of C7 PFHpA [43]. Additionally, other short-chain PFCA compounds, such as C6 perfluorohexanoic acid (PFHxA), C5 per-fluoropentanoic acid (PFPeA), and C4 perfluorobutanoic acid (PFBA), have been detected in the final solution, each with a low concentration of approximately nanomoles per liter [37]. The degradation efficiency in photochemical degradation ranges from 70% to the complete removal of PFOA, with 30% to 90% defluorination occurring during a reaction time of 4–24 h [35]. It is worth noting that the defluorination rate is consistently lower than the PFOA removal rate; Song et al. (2013) achieved up to 100% removal of PFOA within 1 h but only managed a 40% defluorination rate [37]. This phenomenon is due to the higher energy of the C-F bond compared to the C-C bond,

causing the degradation reaction to undergo chain shortening processes before complete defluorination is achieved.

**Table 1.** Photochemical degradation of perfluorooctanoic acid (PFOA).

| Wavelength (nm) | Power | PFOA Initial Concentration (mg/L) | Other Chemical Content | Conditions | Degradation Efficiency | Reference |
|---|---|---|---|---|---|---|
| 254 | 200 W | 140–1387 | 34 mg/L of $H_2O_2$ 0.48 MPa $O_2$ | Room temperature pH 0.8 | 0.674 mg/h Up to100% removal | [26] |
| 220–460 | 200 W | 559 | 9600 mg/L $S_2O_8^{2-}$ | 0.3 pH 0.48 MPa $O_2$ 25 °C | 139.5 mg/h Up to 100% removal | [41] |
| 254 | 15 W | 40 | 500 mg/L b-$Ga_2O_3$ | Anaerobic | 5.3 mg/h 40% removal 15% defloration | [44] |
| 254 | 15 W | 10.35 | 16 mg/L–133 mg/L KI | 9 pH Room temperature anoxic | 1.62 mg/h 93.9% removal 89% defluorination | [35] |
| 254 and 185 | 23 W | 4.14 | 12 mg/L $NaIO_4$ | | 1.8 mg/h 87% removal 25% defluorination | [45] |
| C | 6 W | 50 | 500 mg/L $TiO_2$ | pH4 25 °C | 8.3 mg/h Up to 100% removal 50% defluorination | [39] |
| 254 | 9 W | 8.2 | 1020 mg/L $H_2O_2$ 152 mg/L $FeSO_4$ | 25 °C pH 3 | 1.5 mg/h 58% defluorination 93% removal | [36] |
| 254 | 10 W | 8.3 | 960 mg/L $SO_4^{2-}$ | Anaerobic pH 10.3 | 2.7 mg/h 80% defluorination Up to 100% removal | [37] |
| 254 | 23 W | 19.8 | 1140 mg/L $FeCl_3$ 270 mg/L $H_2C_2O_4$ | Room temperature pH 2.4 anoxic | 3.8 mg/h 80% removal 20% defluorination | [33] |
| 254 | 28 W | 10 | 25 mg/h $O_3$ 1 L/min air 2000 mg/L ZnO | pH 4.5 25 °C | 1.75 mg/h 70% removal | [38] |
| 210–280 | 75 W | 0.11 0.7 L/min | 2 L/min UV ozonated air | | 0.00033 mg/h Up to 100%removal | [46] |
| 254 | 18 W | 10 | 1260 mg/L $Na_2SO_3$ | pH 12 20 °C | 10 mg/h 93% defluorination Up to 100% removal | [47] |
| 350–780 | 500 W | 0.5 | BiOCl/Zn-Al hydrotalcite 500 mg/L | pH 2 | 90% removal 0.075 mg/L | [43] |
| 254 | 23 W | 5 | 2420 mg/L $Fe(NO_3)$ 2000 mg/L $Fe_2(SO_4)_3$ 2550 mg/L $NaNO_3$ | pH2.4 Room temperature Anaerobic | 92% removal | [42] |
| 225–300 | 450 W | 100 | $Zn_{0.5}Cu_{0.5}Fe_2O_4$ 8300 mg/L Oxalic acid 9000 mg/L | pH 2.03 27 °C | 15 mg/h 30% removal | [48] |

### 2.2. Electrochemical Degradation

Electrochemical degradation utilizes boron-doped diamond and $Ti-SnO_2$ electrodes submerged in an electrolyte containing ion-exchange reagents, such as $NaClO_4$, $Na_2SO_4$, $NH_4OH$, and $NH_4Cl$, for PFOA degradation [27,49,50]. This process generates free electrons on the electrode side, which then react with PFOA to produce free fluoride ions and $CO_2$. Table 2 illustrates that the current density during electrical chemical degradation typically varies from 10 to 50 $mA/cm^2$. Higher current densities remove PFOA more quickly but consume more energy [27]. Witt et al. (2020) experimented with various current densities during the degradation process to strike a balance between energy consumption and defluorination performance. During the first hour, they used 50 $mA/cm^2$ to initiate PFOA degradation, then switched to 5 $mA/cm^2$ for long-term defluorination, ultimately resulting in 30% energy savings [49].

As depicted in Table 2, in most of the studies, the reactions occurred at room temperature in a pH 5 acid solution. The optimal electrode separation distance falls within the range of 5–25 mm, dependent on the reacting current density. When the current density remains constant, an increase in the electrode separation distance results in a decrease in the efficiency of the reaction [51]. The degradation of PFOA requires free radicals. Ma et al. (2015) demonstrated that the degradation of PFOA decreased when the pH increased from 5 to 11 or decreased to 3, primarily due to the blocking of radical formation [50]. Table 2 illustrates that most of the electrochemical degradation processes occur in an acidic environment, typically within the pH range of 3 to 5.

However, in certain special conditions, such as in the treatment of wastewater or landfill leachate, the reaction still occurs in a natural environment. Nevertheless, it necessitates a higher current density, and the degradation efficiency is lower compared to the acidic environment [52,53]. A higher initial concentration of PFOA typically leads to an increased reaction rate. However, due to the limitations of ion exchange at the anode, if the initial PFOA concentration surpasses 100 ppm, it can also impede the reaction. This is because elevated levels of PFOA generate numerous short-chain by-products, such as C7 PFHxA, and a higher concentration of these by-products can deplete the available radicals and extinguish the reaction [50].

Similar to photochemical degradation, electrochemical degradation also required 1–4 h to achieve 95% removal of PFOA, along with a 75% defluorination, as reported by Ma et al. in 2015. The degradation products are reminiscent of those produced through photochemical degradation, including short-chain PFCAs like PFHpA, PFHxA, PFPeA, and PFBA. As a result of a stepwise degradation process, intermediate products, such as C7 PFCA, reach their maximum concentrations shortly after the reaction begins. However, as the experiment continues, these intermediate concentrations gradually decrease and are further degraded into even shorter-chain PFCA compounds, such as PHFpA, PFHxA, and PFBA, as documented by Ma et al. in 2015.

### 2.3. Other Degradation Methods

In addition to photochemical and electrochemical degradation methods, a few other techniques are employed for PFOA degradation. These methods include sonochemical, microwave-enhanced, and other chemical-catalyzed degradation methods. Unlike UV light or electrical currents, which serve as energy sources, these chemical degradation processes utilize substances like $Na_2S_2O_8$, argon gas, or vitamin B13 as catalysts. They may combine with various forms of energy or even operate without additional energy input to facilitate degradation.

**Table 2.** Electrical chemical degradation of perfluorooctanoic acid (PFOA).

| Electrodes | Current Density (mA/cm$^2$) | PFOA Initial Concentration (mg/L) | Chemical Reagent | Conditions | Degradation Efficiency | Reference |
|---|---|---|---|---|---|---|
| Ti/SnO$_2$-Sb/PbO$_2$ | 10 | 100 | 10 mg/L NaClO$_4$ | 25 °C pH 5 5 mm distance | 60 mg/h 91% removal 77.4% defluorination | [27] |
| Ti/SnO$_2$-Sb/Yb-PbO$_2$ | 20 | 100 | 12,240 mg/L Na$_2$SO$_4$ | pH 5 25 °C 500 rpm 5 mm distance | 38 mg/h 95% removal 75% defluorination | [50] |
| B and N codoped diamond | 4 | 50 | 6122 mg/L Na$_2$SO$_4$ | pH 4.8 2.5 cm distance | 25 mg/h Up to 100% removal 80% defluorination in 3 h | [54] |
| Boron-doped diamond | 0.6 | 1000 | 12,240 mg/L NaClO$_4$ 1.5% TiO$_2$ | 1 cm distance 600 mW/cm$^2$ UV at 254 nm | 166 mg/h 50% removal | [31] |
| Boron-doped diamond | 21.4 | 5.5 | 1500 mg/L Na$_2$SO$_4$ | 2 cm distance 6.27–8.53 pH | 2.75 mg/h Up to 100% removal 40% defluorination | [52] |
| Boron-doped diamond | 75 | 0.1 | 17.466 mg/L K$_2$HPO$_4$ 250 µL/L H$_3$PO$_4$ | pH 7.03–7.29 20–25 °C 2.5 cm distance | 0.0125 mg/h | [53] |
| Boron-doped diamond | 50 | 1.19 | 6.25% NH$_4$OH 2% CH$_3$OH | Anaerobic | 0.29 mg/h | [49] |
| Ti and boron-doped diamond | 5 V/SHE | 19.8 | 2840 mg/L Na$_2$SO$_4$ | 25 °C 3.5 cm pH 4 anaerobic | Up to 100% removal 90% defluorination 3.96 mg/h | [55] |
| Boron-doped diamond and stainless steel | 10 | 20 | 10% Na$_2$SO$_4$ | pH 3 50 °C 1 cm distance 700 rpm | Up to 100% removal 70% defluorination 10 mg/h | [51] |

In sonochemical degradation, argon gas and ozone are introduced during sonolysis. This process degrades PFOA by generating bubbles under ultrasound. When these bubbles rupture, they create high temperatures at the interface, enabling interface pyrolysis. Argon gas, characterized by a high polytopic index, exhibits greater heat flux during transitions between different thermodynamic states. Consequently, the addition of argon gas during degradation can elevate the interface temperature and accelerate the reaction rate [28]. Sonochemical degradation employs a 200–250 W ultrasound generator operating at a frequency of 200–612 kHz to facilitate a first-order reaction with a rate of 0.032 min$^{-1}$ [28,56]

As shown in Table 3, the reaction temperature typically falls within the range of 10–20 °C, and the reaction occurs in an acidic solution. However, this method has several limitations. Firstly, it can only effectively degrade PFOA at low concentrations (10–100 ppb of PFOA). Secondly, when treating contaminated water sources such as landfill groundwater, it can be susceptible to the influence of volatile compounds like acetone, diisopropyl ether, and butanone, as these compounds can lower the interfacial temperature and hinder the formation of OH radicals [56].

Researchers have also observed that microwaves enhance degradation in sonochemical processes. In the study of Li et al. (2021), a 300 W microwave oven was used to generate a 2450 MHz microwave, heating specific compounds. This reaction, conducted under high temperatures in an acidic environment, produced free radicals, resulting in the removal of

up to 80% of PFOA in just 4 h [30]. During sonochemical degradation, chemicals such as $H_2O_2$ are added to initiate Fenton-like processes or $Na_2S_2O_8$ to produce $SO_4^-$ ions, which then engage in persulfate oxidation reactions [30,57]. However, similar to photochemical reactions, the degradation rate is significantly affected by pH and temperature and can also be hampered by the presence of Cl- ions [30]. Another notable drawback of this reaction is its high energy consumption, approximately 458 kWh/mol, owing to the dissipation of energy as heat during the reaction [29].

Some chemical degradation processes do not require additional power input to initiate degradation. For instance, [29] introduced a method that employs vitamin B13 as an electron transporter in combination with CuSO4 and NaBH4, resulting in the formation of zero-valence copper to expedite the reaction [29]. By adding 45 nM of titanium citrate as a reducing agent, they achieved a 65% removal of PFOA within 18 h [29]. In another approach, researchers utilized KMnO4 in combination with HCl for PFOA degradation. However, this method, although effective, required an extended period of up to 6 months to remove 90% of contaminants [58]. Comparing the experiment of Fang et al. (2016) with other chemical degradation methods reveals that even without the need for additional power input, degradation can occur. Nevertheless, due to the stubborn stability of PFOA, the reaction rate remains relatively low. Balancing energy consumption with degradation efficiency is a potential avenue for future research in chemical degradation.

**Table 3.** Other chemical degradation experiments of perfluorooctanoic acid (PFOA).

| Catalytic Reagent | PFOA Initial Concentration (mg/L) | Chemical Supplement | Conditions | Degradation Efficiency | Reference |
|---|---|---|---|---|---|
| Ultrasound 200 kHz 3 w/cm$^2$ | 10 | Ar | 20 °C Anaerobic pH 4.8–3.5 | 8.5 mg/h 85% removal | [28] |
| Ultrasound 354–612 kHz | 0.1 | Ar | 10 °C Anaerobic | 0.028 mg/h 56% removal | [55] |
| Ion-modified diatomite | 10 | 68 g/L $H_2O_2$ 1 g modified diatomite | 25 °C pH 9 | 1.38 mg/h | [31] |
| 800 W microwave 2450 MHz | 105 | 10 mM $S_2O_8^{2-}$ | pH 2.5 90 °C | 22 mg/h 85% removal | [30] |
| | 100 | 0.1% KMnO$_4$ and 0.36% HCl | 24 °C Domestic light Shaking 1 per day | 0.5 mg/L/day 90% removal | [58] |
| Vitamin B12 Copper nanoparticles | 50 | 2 g/L copper dose 0.2 mM vitamin B12 45 mM Titanium Citrate | pH 9 70 °C 100 rpm Anoxic | 1.35 mg/h 65% removal | [29] |
| 300 W Microwave Pb-BiFeO$_3$/rGO | 50 | 44 mg/L $H_2O_2$ 1 g/L Pb-BiFeO$_3$/rGO | pH 5 90 °C | 9 mg/min 90% removal | [57] |

## 3. Biological Degradation

Biological degradation represents an emerging technology when compared to chemical degradation, and research in this field remains rather limited. Only a handful of research articles have been published, with a specific focus on the biodegradation of PFOA. Previous research has demonstrated the thermodynamic favourability of dehalogenation reactions and the maturity of microbial dechlorination technology. This indicates that bacteria can indeed derive energy from breaking down perfluoro compounds [6]. However, the majority of biodegradation research centers on the degradation of fluorotelomer alcohols (FTOH), which ultimately yield PFOA as a final product without further breakdown [6,59].

Furthermore, it has been reported that biodegradation typically requires at least one hydrogen atom to be attached to the alkyl chain to initiate the reaction. In the case of PFOA, however, the alkyl chain only features fluorine atoms, forming high-energy C-F bonds (485 kJ/mol), which render them even more challenging to degrade compared to other perfluorinated compounds, such as fluorotelomer alcohols [60].

Schröder et al. (2004) employed sewage-treatment-plant sludge as a bacterial seed in a closed-loop system to treat PFOA for 28 days [61,62]. In recent years, scientists have managed to isolate and acclimate Pseudomonas Parafulv and Acidimicrobium sp A6 strains from PFOA-enriched environments, allowing them to perform biological degradation reactions using PFOA as a carbon source. Bacteria exhibit a certain level of tolerance to PFOA concentrations; when the concentration of PFOA reaches 500 ppm, the bacterial strain reaches a maximum population of 0.175 OD600 and achieves a degradation rate of 30% over 72 h [2]. However, if the concentration exceeds this threshold, both the OD600 and degradation rate begin to decline due to the inhibitory effects of PFOA on bacterial growth.

In Table 4, biodegradation can occur either aerobically or anaerobically under culturing conditions of 30 °C and 150 rpm. However, the majority of experiments are conducted in anaerobic environments [63]. The primary reason for biological degradation occurring in an anaerobic environment rather than an aerobic one is the highly oxidized nature of PFOA. It struggles to donate electrons for further oxidation reactions in an aerobic environment [22]. In anaerobic conditions, PFOA can preferentially accept electrons to undergo reduction reactions, leading to no removal of PFOA under Liou's aerobic degradation conditions but achieving 67% removal in Huang's anaerobic degradation experiment [63]. Table 4 presents the anaerobic degradation of PFOA carried out by an Acidimicrobium sp A6 strain in an acidic environment with a pH range of 4.5–5.5. There has only one aerobic experiment, conducted by Yi et al. (2016), in a neutral environment using Pseudomonas parafulva for degradation. During their experiment, Yi et al. (2016) supplemented yeast extract and glucose to support metabolism. This can be used during the acclimation and isolation stages to boost bacterial populations during the degradation processes and increase the final removal rate of PFOA [2].

**Table 4.** Biological degradation of perfluorooctanoic acid (PFOA).

| Sources of Bacteria | Initial PFOA Concentration | Reaction Environment | Chemical Supplement | Degradation Efficiency | Reference |
|---|---|---|---|---|---|
| *Pseudomonas parafulva* | 500 mg/L | pH 7<br>30 °C<br>160 rpm | 1000 mg/L yeast extract<br>2% inoculum<br>5000 mg/L $NH_4NO_3$<br>2000 mg/L NaCl<br>1000 mg/L $KH_2PO_4$<br>1000 mg/L $K_2HPO_4$<br>500 g/L $MgSO_4 \cdot 7H_2O$<br>50 mg/L $CaCl_2 \cdot 2H_2O$ | 48% removal<br>In 5 day | [2] |
| *Acidimicrobium* sp. Strain A6 | 0.1 mg/L | Anaerobic<br>4.5 pH<br>30 °C<br>150 rpm | 1260 mg/L $Fe_2O_3 \cdot 0.5H_2O$<br>150 mg/L $NH_4Cl$<br>25 mg/L $(NH_4)_2SO_4$<br>20 mg/L $NaHCO_3$<br>71 mg/L $KHCO_3$<br>7.523 mg/L $KH_2PO_4$<br>101 mg/L $MgSO_4 \cdot 7H_2O$<br>58.8 mg/L $CaCl_2 \cdot 2H_2O$<br>1 mL trace element<br>1 mL vitamin solution | 60% removal<br>In 100 d | [63] |

**Table 4.** *Cont.*

| Sources of Bacteria | Initial PFOA Concentration | Reaction Environment | Chemical Supplement | Degradation Efficiency | Reference |
|---|---|---|---|---|---|
| *Acidimicrobium* sp. Strain A6 | 10 mg/L | 25 °C Anaerobic pH 4.5–5 | 506 mg/L $Fe_2O_3 \cdot 0.5H_2O$<br>177 mg/L $NH_4Cl$<br>77.9 mg/L $(NH_4)_2SO_4$<br>19.8 mg/L $NaHCO_3$<br>71 mg/L $KHCO_3$<br>9 mg/L $KH_2PO_4$<br>100 mg/L $MgSO_3 \cdot 7H_2O$<br>60 mg/L$CaCl_2 \cdot 2H_2O$<br>1 mg/L trace element<br>1 mL/L vitamin solution | 67.7% removal in 150 d | [64] |
| *Acidimicrobium* sp. Strain A6 | 100 mg/L | Anaerobic pH 5–5.5 240 rpm Room temperature stainless steel as cathode and graphite plate as anode. | 203.3 mg/L $NH_4Cl$<br>79.28 mg/L $(NH_4)_2PO_4$<br>20.16 mg/L $NaHCO_3$<br>71 mg/L $KHCO_3$<br>8.98 mg/L $KH_2PO_4$<br>101 mg/L $MgSO_4 \cdot 7H_2O$<br>45.5 mg/L $CaCl_2$<br>55.24 mg/L AQDS<br>1 mL/L vitamin supplement | 77% removal in 18 d | [60] |

The most significant degradation occurs in an anaerobic environment using a closed-loop system for treating PFOA-containing wastewater, achieving up to 100% removal within 28 days [61]. However, it does not indicate the defluorination rate during the reaction. Recent research employed Acidimicrobium sp A6 Strain for anaerobic degradation, achieving a maximum defluorination rate of 63% over a 100-day reaction period [63]. When researchers substituted iron with electrolysis cells, similar degradation results were achieved, removing 60–70% of PFOA within a shorter timeframe of 18 days [60]. Nevertheless, all these experiments were conducted on a laboratory scale in a monoculturing environment and were not directly used for wastewater treatment.

When the initial PFOA concentration increases from 0.1 to 200 ppm, the consumption rate of the reduction reagent does not change significantly. This indicates that the reaction rate is barely affected by PFOA at low initial concentrations [63]. In aerobic reactions, when the concentration increases from 0 to 500 ppm, the bacterial population also increases, showing that higher PFOA levels can stimulate bacterial growth. However, due to the toxicity of PFOA, microbacteria exhibit a certain tolerance to PFOA concentrations in the culturing media. When the concentration of PFOA exceeds 500 ppm, the reaction rate decreases because the toxicity of PFOA inhibits bacterial growth [2].

To maintain proper bacterial growth in a monoculture environment, it is necessary to add other chemical compounds to the culturing solution. Carbon, nitrogen, and phosphorus are the primary elements required by bacteria for building their cells. Therefore, studies have utilized $NH_4Cl$, $(NH4)_2SO_4$, and $NH_4NO_3$ as nitrogen sources and $KH_2PO_4$ and $K_2HPO_4$ as phosphate sources [64,65]. Regarding the carbon source, some studies used PFOA, intending for bacteria to consume it as an energy source during metabolism. However, adding glucose as a co-metabolism compound can also stimulate bacterial growth and increase the degradation rate of PFOA from 30% to 45% [2]. Calcium and magnesium are essential elements for maintaining proper bacterial metabolism, and they have been added in the form of $MgSO_4$ and $CaCl_2$. Additionally, other trace elements and vitamin solutions have been introduced into this reaction. Lastly, alkalinity can act as a buffer for long-term culturing periods. Therefore, most of the experiments shown in Table 4 included $NaHCO_3$ and $KHCO_3$ as additional alkalinity sources.

## 4. Comparison between Biological Degradation and Chemical Degradation

The chemical degradation of PFOA has been extensively developed, with numerous experimental studies conducted using various degradation methods. In contrast, biological degradation represents a frontier technology, with only around six published papers available online. Limited resources and data further compound the challenge of comparing chemical and biological degradation in terms of degradation products, degradation mechanisms, and degradation efficiency.

### 4.1. Degradation Products

During chemical degradation, three main types of degradation products are typically observed: firstly, free fluoride ions; secondly, $CO_2$; and thirdly, short-chain PFCA compounds such as PFHpA, PFHxA, PFPeA, and PFBA. The final concentration of PFOA and other short-chain PFCA degradation products is determined using liquid chromatography–mass spectrometry (LC-MS), while ion chromatography (IC) is employed as the testing method for fluoride ions [63].

In previous studies s the concentration of PFOA has decreased, the levels of degradation products and by-products, such as free fluoride and short-chain PFCAs, has increased simultaneously [36]. The concentration of the degradation by-product, especially C7 PFHpA, typically reaches a peak relatively quickly because there is a sufficient amount of PFOA available to undergo a chain-shortening process to form PFHpA. Moreover, PFHpA exhibits a lower decomposition rate compared to other shorter-chain PFCAs [65]. However, after PFHpA reaches its peak, its concentration decreases significantly over the remaining reaction time to form other shorter-chain PFCAs [33,35]. This is due to the fact that, at room temperature, the reaction rate of C7 to C6 PFCA ($8 \times 10^{10}$ s$^{-1}$) is lower than that of C6 to C5 PFCA ($2 \times 10^{11}$ s$^{-1}$). Therefore, after C8 PFOA transforms into C7 PFHpA, it accumulates for a while before undergoing further degradation [66].

Table 1 provides a summary of previous studies that have achieved 100% removal of PFOA. However, none of these studies has been able to achieve 100% defluorination, meaning that short-chain PFCA degradation by-products still exist in the solution. In the final solution, monitoring the fluoride ion is crucial, as it can be used to calculate the fluorine balance and evaluate the defluorination percentage [63]. Other short-chain perfluorocarboxylic acids, such as 7C PFHpA, 6C PFHxA, and 5C PFBA, are also present in the final solution [37]. These short-chain PFCAs are the end products of incomplete defluorination, and their concentrations can vary depending on different reaction methods and reaction times. When the defluorination rate is higher, shorter PFCA by-products are more likely to be detected, indicating that more alkyl chains have been broken down and more fluoride ions have been released [35].

Compared to chemical degradation, similar degradation products are detected in biological degradation, including free fluoride and short-chain PFCA. In all four experiments presented in Table 4, short-chain PFCA compounds like PFHxA, PFHpA, PFPeA, and PFBA were observed after a period of culturing. Additionally, free fluoride ions have been detected as evidence of PFOA degradation. As the concentration of PFOA decreases during the reaction, the concentration of free fluoride ions increases, demonstrating a phenomenon similar to chemical degradation [60]. After a 150-day degradation period, PFHpA is found in a higher proportion in the final solution compared to other short-chain compounds [63]. This could be attributed to the poor biodegradability of PFOA; it can only undergo the initial steps of chain shortening processes during biodegradation and rarely progresses further, unlike chemical degradation [63].

Overall, both biological and chemical degradation yield similar degradation products, and PFOA undergoes a comparable stepwise degradation process. These short-chain PFCA products and intermediate byproducts highlight the persistence of PFCA. Therefore, it is essential to focus not only on the reduced concentration of PFOA but, more importantly, on the defluorination rate. This is crucial because these short-chain PFCA products can still pose a significant concern for overall PFCA contaminants in the environment. In real

conditions for treating highly contaminated wastewaters, PFOA is removed during the process. However, at the same time, other byproducts are produced, potentially causing an increase in PFOA concentration in the treated solution [67]. Therefore, even though current technology can successfully remove PFOA on a laboratory scale, it remains a substantial challenge to apply in real industrial settings.

### 4.2. Degradation Mechanism

The degradation intermediate product demonstrates that PFOA undergoes degradation in a stepwise manner. The reaction environment indicates that PFOA can be degraded through oxidation and reduction reactions. The reduction reaction occurs in an anaerobic alkaline environment, generating free electrons from $I^-$, $SO_3^{2-}$, or $Ti^-$ with energy input [35,37,63]. In Figure 2, it is illustrated how free electrons attack the fluorine atom at the alpha position on the alkyl chain, as it is the most susceptible site for attack [37]. Subsequently, it replaces fluorine with hydrogen one after another. Once both fluorines have been replaced by hydrogen, the C-C bond on the alkyl chain breaks, releasing CH2 and shortening the alkyl chain.

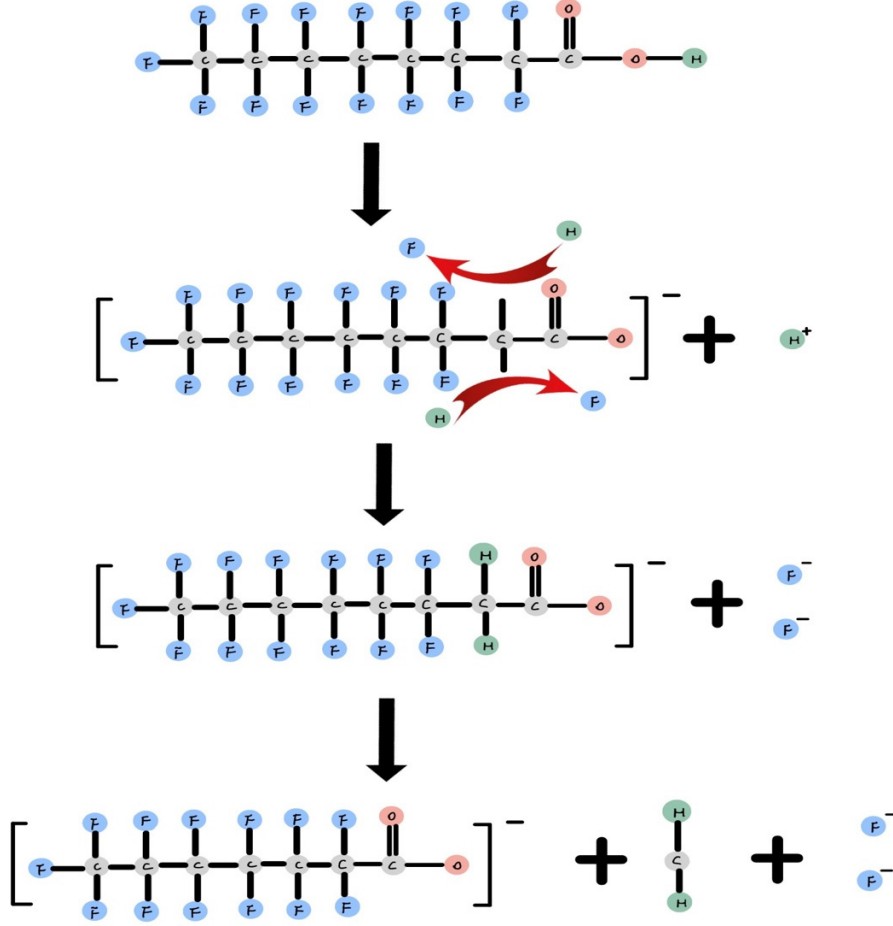

**Figure 2.** Oxidation reaction of PFOA [37].

Figure 3 shows during an oxidation reaction, $H_2O_2$ and $O_3$ are used in combination with ferric ions or persulfate in an aerobic environment to generate hydroxyl radicals (OH•). When these OH• radicals attach to dissolved PFOA, they remove the carboxylic group as $CO_2$, forming a C7F15 radical [36]. Subsequently, a water molecule attaches to the radical, facilitating hydroxylation and the elimination of one fluorine atom. Afterward, another water molecule is involved in hydrolysis, releasing another fluoride ion [9]. Following the elimination of fluoride ions and the shortening of the alkyl chain, C8 PFOA transforms into

C7 PFHpA, representing the initial stage of intermediate product degradation. This process continues, gradually breaking down C7 PFHpA into C6 PFHxA, and further into C5 PFPeA as the alkyl chain becomes shorter. This degradation process ultimately results in the formation of fluoride ions and $CO_2$ [65]. Despite PFOA's inherent stability, Gomez-Ruiz's photocatalytic degradation experiment reduced the total organic carbon (TOC) by 62%, indicating mineralization during PFOA degradation [65].

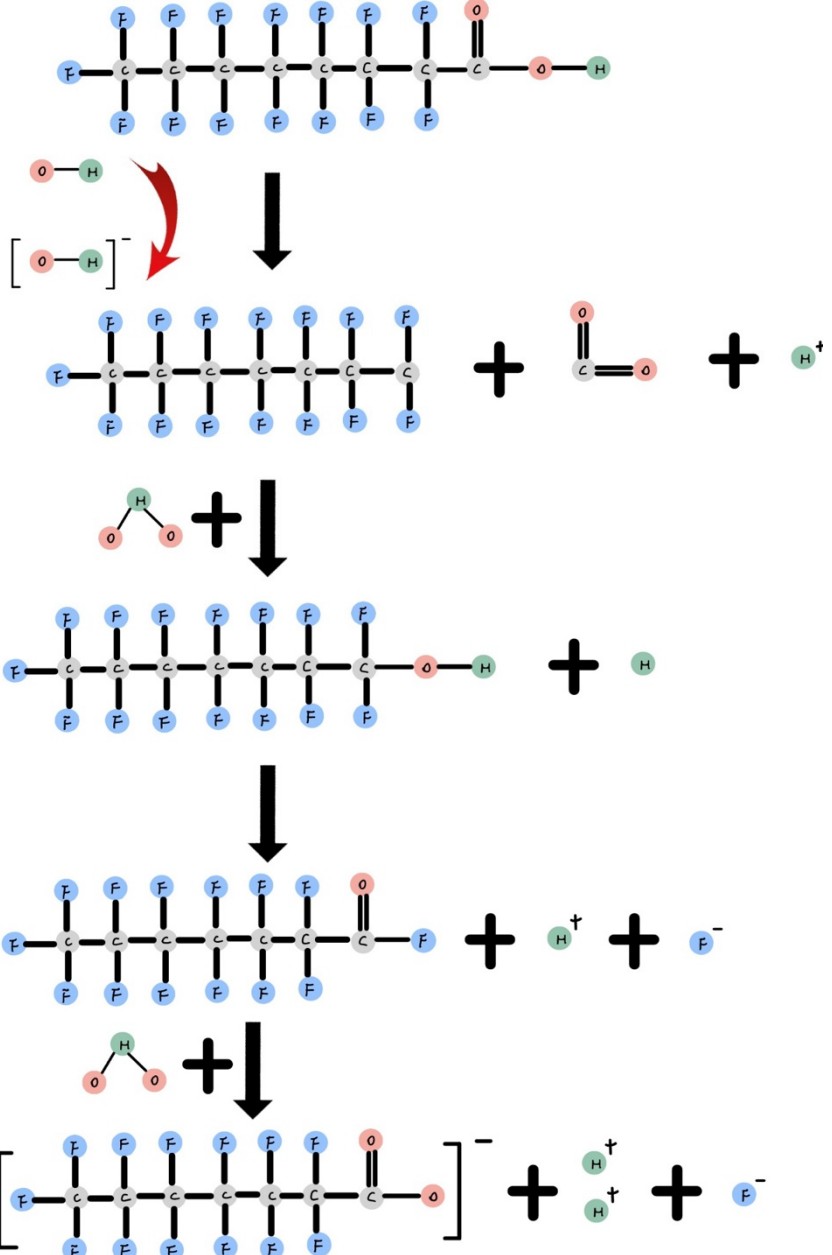

**Figure 3.** Reduction reaction of PFOA.

Biodegradation can occur in both aerobic and anaerobic environments, producing similar degradation products as chemical degradation. However, there is limited research available on biodegradation mechanisms, and the metabolic processes are more complex than chemical reactions. Huang et al. (2019) employed ammonium as an ion contributor and iron oxide as an ion acceptor to drive the Feammox process using Acidimicrobiaceae sp. strain A6. Researchers discovered a novel reductive dehalogenase in this bacterial strain, enabling simultaneous electron transfer to iron and PFOA for degradation [63].

Ruiz-Uriguen et al. (2022) also proposed a stepwise biodegradation mechanism based on the concentration of intermediates produced during the degradation of A6 bacteria [60].

### 4.3. Compare the Degradation Efficiency between Chemical and Biological Methods

When evaluating degradation efficiency, two primary standards are typically considered. The first standard involves assessing the PFOA removal rate in comparison to the initial concentration, while the second standard is based on defluorination, which is calculated by monitoring the final concentration of $F^-$ ions in the solution. Several studies listed in Tables 1–3 demonstrate that chemical degradation generally takes several hours to achieve 90% degradation with 30–90% defluorination [53]. The efficiency of the reaction can vary depending on the specific chemical degradation methods and power inputs used, but the degradation process typically occurs within less than a day. In comparison, when looking at the biological degradation experiments presented in Table 4, it becomes apparent that the biological degradation method exhibits lower degradation efficiency, achieving a 48–70% removal of PFOA but requiring a significantly longer period (around 100 days) to achieve a similar level of degradation [2,62]. Tables 1–4 also indicates that biological degradation typically deals with higher PFCA concentrations (100–500 ppm) when compared to chemical degradation (4–100 ppm). This is because, in biological degradation, bacteria utilize PFOA as a carbon source for their metabolism, which necessitates higher initial concentrations for consumption. Another reason for the lower removal percentage in biological degradation is the higher initial concentration that these methods typically start with.

In the context of chemical degradation, due to the first-order kinetic reaction, the concentration of PFOA decreases significantly when no other competing compounds are present. In the experiment conducted by Song et al. (2013), for instance, during a 24 h degradation period, the first hour of the reaction removed approximately 80% of PFOA, and C7 PFHpA also exhibited a noticeable peak simultaneously [37]. After reaching their maximum concentrations, these intermediates also decrease significantly, following the first-order reaction rate, similar to what is observed with PFOA. In another experiment by Qu et al. (2010), short-chain intermediate PFCA decreased by around 60% within the first 4 h of reaction time but only decreased by 20% in the subsequent 10 h [35].

In the context of biological degradation, the concentration of PFOA did not significantly decrease at the beginning of the reaction due to the acclimation and growth of bacteria during the lag phase and log phase. Huang et al. (2019) discovered that the PFOA concentration decreased by approximately 50% over 30 days, with a consistent rate of reduction, indicating that it does not follow a first-order reaction [63]. Moreover, short-chain-degradation intermediate products, such as PFHpA, PFHxA, and PFBA, gradually reached their maximum concentration, which was around 1/3 or 1/2 of the total reaction time, without displaying a distinct peak [63]. This suggests that biological degradation proceeds at a slower reaction rate, and neither PFOA nor the short-chain intermediate products adhere to a first-order reaction, unlike chemical degradation. Once degradation intermediates are produced, they react immediately and do not accumulate in the solution over time.

The reaction rate can be influenced by numerous factors. During electrochemical degradation, an increase in the initial concentration from 10 ppm to 100 ppm results in an increase in the reaction rate, but when the concentration surpasses 200 ppm, the reaction rate decreases. The presence of reaction intermediates such as PFHpA blocks the limited reaction surface and inhibits the further degradation of PFOA [36]. In the case of aerobic biodegradation, an increase in the initial concentration from 100 ppm to 500 ppm leads to an increase in the degradation rate, from 25% to 33%. However, when the concentration exceeds 500 ppm, the degradation rate drops to 10% due to the toxicity of PFOA, which hinders the growth of aerobic bacteria [2]. The presence of other contaminants in the solution can also affect the degradation processes. Given the high stability of perfluoro compounds, if other competing compounds, such as organic or nitrogen compounds, are

present, the reaction tends to prioritize these other compounds over PFOA degradation. Adding 10 mM of $NO_3$ to the solution during photochemical degradation reduces the defluorination efficiency at 24 h from 80% to 2.3% [37]. Considering these observations, in the case of highly contaminated wastewater, such as landfill leachate, chemical degradation of PFOA may not occur, as the reaction tends to target other contaminants first. Additionally, if other PFCs are present in the reaction solution, the reaction will initially occur with these less stable PFCs, ultimately leading to more PFOA as a degradation product [9].

Current research indicates that the biodegradation rate of PFOA is much lower than chemical degradation. However, further research could investigate the detailed mechanisms that occur at the cellular level and determine which enzymes are involved. Recent studies have shown that a plant protein called Cannabis Sativa L can remove PFOS and PFHxS up to 89% within one hour [68]. Additionally, combining chemical degradation with other methods may increase reaction efficiency. For instance, when using an Acidimicrobium sp. A6 strain along with conventional chemicals as electron donors and acceptors, it took 30 days to achieve 60% degradation. However, when employing microbial electrolysis cells, only 18 days were needed to achieve up to 77% degradation [60]. These findings demonstrate that continuous research efforts are improving the reaction environment and delving deeper into the reaction mechanisms. As a result, biological degradation still holds significant promise for further development.

## 5. Future Outlook

In recent years, PFOA has garnered increased attention as an emerging contaminant. It has been detected in various environmental matrices, ranging from soil, water, and sediment to wastewater. While several successful degradation experiments have been conducted on a laboratory scale, most of these are carried out on spiked samples with high concentrations (in the ppm range). PFOA's typical concentration in natural environments falls within the ppb range, making it challenging to degrade. Moreover, other contaminants in the environment often have higher concentrations and lower stability, which hinders the degradation of PFOA. As PFOA undergoes stepwise degradation, its transformation products, primarily other short-chain PFCAs, may still persist in the solution, exhibiting similar toxicity. Additionally, the high energy consumption during chemical reactions can make scaling up to an industrial level economically unviable.

When it comes to biodegradation, despite its potential as a cutting-edge technology, research in this area remains limited. It offers numerous advantages compared to chemical degradation, such as energy efficiency and the absence of harmful by-products from degradation. However, drawbacks are evident as well. Firstly, the degradation process is time-consuming, requiring 60–100 days to complete. This lengthy period can pose challenges for wastewater treatment plants that need to hold wastewater for extended periods before discharge. For bacteria to effectively consume PFOA during metabolism, they require a high concentration, typically around 100 ppm, which is significantly higher than the concentration found in wastewater. Consequently, it does not make practical sense to spike a contaminant 10,000 times higher and then only achieve a 60–70% removal in real-life scenarios. Moreover, the A6 bacteria strain only demonstrates degradation under anaerobic conditions in monoculture environments, which is not applicable in real-world wastewater settings.

Most PFOA chemical degradation has been reported using advanced oxidation processes (AOP) and photochemical degradation methods. These degradation methods are akin to conventional wastewater treatment processes. In a wastewater treatment plant, some PFOA can still be simultaneously degraded. Researchers should also focus their treatment efforts on pollution sources such as landfill sites, industrial wastewater treatment facilities, and firefighting stations, rather than natural water systems. This is because, at the source of pollution, the concentration of contaminants is often higher, making degradation easier. Low PFOA concentrations could prove challenging to degrade when targeted within a natural water system. Single methods alone may not promise efficient PFOA

treatment; instead, hybrid methods are required. When dealing with contaminants with high concentrations, the initial approach may involve applying biodegradation to reduce the concentration within the elevated range. Subsequently, a chemical method can be employed as a secondary treatment to further degrade contaminants at lower concentrations. However, considering the current drinking water standard in Canada, which is measured in parts per trillion (ppt), no degradation methods are available for such low concentrations. Therefore, the only viable option is to use conventional physical removal methods, such as employing a GAC absorbent as a tertiary treatment method.

Nonetheless, chemical or biological methods can be utilized for treatment once contaminants have been concentrated in the absorption media. This approach allows for more efficient treatment of contaminants when they are present in high concentrations after being concentrated in one location. It also enables the recycling of the absorbent media, making the overall process more economically feasible. Additionally, the tertiary treatment eliminates trace contaminants and effectively removes toxic by-products, which is crucial for ensuring water safety.

### 6. Conclusions

PFOA is one of the main degradation products of other perfluoro compounds. Researchers employ physical methods to remove it and utilize chemical or biological methods for its decomposition. The chemical degradation method involves applying energy input along with the presence of chemical catalysts to perform oxidation or reduction reactions on PFOA. This method exhibits higher efficiency and typically achieves a high removal rate of PFOA, up to 100% within 24 h. It can also attain a high defluorination rate of up to 98%. Biological degradation is a more cutting-edge technology with limited research and experiments conducted on it. The reaction period for biological degradation typically takes around 100 days to achieve the same degradation performance as chemical degradation, which results in approximately 60% removal of PFOA. Both methods yield similar degradation products, such as fluoride ions and short-chain PFCA. The reaction reagents and final products indicate that the degradation mechanisms between chemical and biological degradation can be similar. Oxidation reactions occur in an acidic environment, while reduction reactions occur in an anaerobic, basic environment. The presence of any type of competitive compound or changes in the reaction environment can significantly affect the degradation performance. Chemical degradation exhibits advantages in many aspects as a matured technology, whereas research on biological degradation is only beginning. With the development of more genetic sequencing testing mechanisms, there is an increased opportunity for researchers to identify new bacterial species capable of using PFOA as an energy source, leading to higher removal efficiency.

**Author Contributions:** X.S.: Conceptualization, Methodology, Investigation. R.P.: Conceptualization, Writing—review & editing, Supervision. P.K.: Writing—review & editing, Supervision. S.K.B.: Supervision, Project administration, Funding acquisition. All authors have read and agreed to the published version of the manuscript.

**Funding:** The authors are sincerely thankful to the Natural Sciences and Engineering Research Council of Canada (Discovery Grant 355254, CRD Grant, Environment and Climate Change Canada; and Alliance Grant 447075) for financial support.

**Institutional Review Board Statement:** Not applicable.

**Informed Consent Statement:** Not applicable.

**Data Availability Statement:** No new data were created or analyzed in this study. Data sharing is not applicable to this article.

**Acknowledgments:** The support of James and Joanne Love Chair in Environmental Engineering at York University is also appreciated.

**Conflicts of Interest:** The authors declare no conflict of interest.

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
