# Peer review of "Comparison between Chemical and Biological Degradation Processes for Perfluorooctanoic Acid"

_soilsystems, doi:10.3390/soilsystems7040091_

Round 1

Reviewer 1 Report

Manuscript No: soilsystems-2564599

Title : Comparison between chemical and biological degradation  processes for perfluorooctanoic acid

Journal : Soil Systems

In this paper, the authors report a review on the comparison between chemical and biological degradation  processes for PFOA. In general, the topic of the review is timely and of interest to scientists working in water treatment related fields. However, there are a few technical issues that need to be addressed to be suitable for publication. My specific comments in order to improve the manuscript are given as follows:

1)     It would be better if the authors provide a graphical abstract.

2)     Keywords can be revised with providing the most identifying ones.

3)     The abstract should emphasize the novelty, the principal results and major conclusions of the investigated topic.

4)     The introduction should clearly show the difference between previous works and the present review related to the chemical and biological degradation  processes for PFOA. The novelty of the presented review must be clearly addressed.

5)     The number of cited references can be enhanced for an effective review paper. Literature survey can be improved related with discussion on the past studies on the topic, the methods, the importance and the applications.  It would be better to add some additional related papers on water treatment techniques, with a few recommendations that would be helpful for readers' understanding the topic:

Research Progress on Process-Intensified Water Treatment Applications, Separations.  2022; 9(11), 353. https://doi.org/10.3390/separations9110353.

6)     Please check the typing errors in the manuscript.

In my opinion, the manuscript should be revised  taking into account the above comments,  before consideration for publication in Soil Systems.  

   Please check the typing errors in the manuscript.

Author Response

Response to Reviewers

We thank all the reviewers for their valuable comments and suggestions. All the comments have been addressed, and the manuscript has been revised suitably. The changes are highlighted in yellow in the revised manuscript.

Response to Reviewer 1

In this paper, the authors report a review of the comparison between chemical and biological degradation processes for PFOA. In general, the topic of the review is timely and of interest to scientists working in water treatment-related fields. However, there are a few technical issues that need to be addressed to be suitable for publication. My specific comments in order to improve the manuscript are given as follows:

Comment#1 It would be better if the authors provide a graphical abstract.

Response: The authors would like to thank you for the reviewer comment and the graphical abstract has been included in the revised manuscript (Added on line 24).

Comment#2 Keywords can be revised with providing the most identifying ones.

Response: The authors would like to thank you for the reviewer comment and the keywords have been updated in the revised manuscript.  (Updated in line 27).

Comment#3 The abstract should emphasize the novelty, the principal results and major conclusions of the investigated topic.

Response: The authors would like to thank you for the reviewer's comment and the abstract has been revised in the revised manuscript (Added in lines 19-24).

Comment#4    The introduction should clearly show the difference between previous works and the present review related to the chemical and biological degradation  processes for PFOA. The novelty of the presented review must be clearly addressed.

Response: The introduction has been revised and the the required statements have been added under the section “Research Gap and Innovations” ( Added lines 117-128).

Comment#5 The number of cited references can be enhanced for an effective review paper. Literature survey can be improved related with discussion on the past studies on the topic, the methods, the importance and the applications.  It would be better to add some additional related papers on water treatment techniques, with a few recommendations that would be helpful for readers' understanding the topic:

Research Progress on Process-Intensified Water Treatment Applications, Separations.  2022; 9(11), 353. https://doi.org/10.3390/separations9110353.

Response: The authors would like to thank you for the reviewer comment and new references have been updated in line 94, 120 -122.

Comment#6  Please check the typing errors in the manuscript.

Response: The authors would like to thank you for the reviewer comment and the manuscript has been revised accordingly.

Comment#7 In my opinion, the manuscript should be revised taking into account the above comments, before consideration for publication in Soil Systems.  

Response: The authors would like to thank you for the reviewer’s comment

Reviewer 2 Report

Comments:

0. Major revision. 1. The novelty of this study should be inserted in the text clearly. 2. The advantages and disadvantages of this study should be investigated. 3. How does your paper contribute to the advancement of knowledge? 4. What are the gap areas and the new contribution in the paper? 5. The stability of the photocatalysts after the degradation processes should be added. 6.  The photocatalyst regeneration should be added. 7.  The degradation mechanism should be studied in detail. 8. The pollutant degradation kinetics should be studied. 9. The conclusion should be improved with clear quantitative findings. 10. The manuscript is satisfactory, however, a careful check is needed. 11. The “introduction” section of the manuscript can be strengthened and supported with some papers related to the literature and cited (optional for authors): Composites Part B: Engineering 184 (2020), 107666; Journal of environmental chemical engineering 5 (2017), 3684-3689.    

Comments:

0. Major revision. 1. The novelty of this study should be inserted in the text clearly. 2. The advantages and disadvantages of this study should be investigated. 3. How does your paper contribute to the advancement of knowledge? 4. What are the gap areas and the new contribution in the paper? 5. The stability of the photocatalysts after the degradation processes should be added. 6.  The photocatalyst regeneration should be added. 7.  The degradation mechanism should be studied in detail. 8. The pollutant degradation kinetics should be studied. 9. The conclusion should be improved with clear quantitative findings. 10. The manuscript is satisfactory, however, a careful check is needed. 11. The “introduction” section of the manuscript can be strengthened and supported with some papers related to the literature and cited (optional for authors): Composites Part B: Engineering 184 (2020), 107666; Journal of environmental chemical engineering 5 (2017), 3684-3689.    

Author Response

Response to Reviewers

We thank all the reviewers for their valuable comments and suggestions. All the comments have been addressed, and the manuscript has been revised suitably. The changes are highlighted in yellow in the revised manuscript.

Response to Reviewer 2

Comments:

Comment#1. Major revision. 1. The novelty of this study should be inserted in the text clearly.

Response: The authors would like to thank you for the reviewer's comment and the manuscript has been revised accordingly (Added in line 115-126).

Comment#2. The advantages and disadvantages of this study should be investigated.

Response: The authors would like to thank you for the reviewer's comment and the manuscript has been revised accordingly (Added in line 505 -524).

Comment#3. How does your paper contribute to the advancement of knowledge

Response: The authors would like to thank you for the reviewer's comment and the manuscript has been revised accordingly (Added in line 122-126).

Comment#4. What are the gap areas and the new contribution in the paper?

Response: The research gaps and the new contribution has been included in the revised accordingly (Added in line 116 to 121).

Comment#5. The stability of the photocatalysts after the degradation processes should be added.

Response: As per the reviewer’s comment, the stability of the photocatalyst has been included in the revised manuscript (Updated in line 170-181).

Comment#6.  The photocatalyst regeneration should be added.

Response: As per the reviewer’s comment, the photocatalyst regeneration has been included in the revised manuscript. (updated in line 170-181).

Comment#7.  The degradation mechanism should be studied in detail.

Response: As per the reviewer’s comment, the degradation mechanism has been included in the revised manuscript (It has been mentioned in section 4.2 line 407 -446).

Comment#8. The pollutant degradation kinetics should be studied.

Response: As per the reviewer’s comment, the pollutant degradation kinetics has been included in the revised manuscript (Added in line 356-358, 414 -418).

Comment#9. The conclusion should be improved with clear quantitative findings. 

Response: As per the reviewer’s comment, the conclusion has been revised and updated in the revised manuscript (Detail quantitative data updated in line 553-555).

Comment#10. The manuscript is satisfactory, however, a careful check is needed.

Response: The authors would like to thank you for the reviewer’s comment and the manuscript has been revised carefully.

Comment#11. The “introduction” section of the manuscript can be strengthened and supported with some papers related to the literature and cited (optional for authors): Composites Part B: Engineering 184 (2020), 107666; Journal of environmental chemical engineering 5 (2017), 3684-3689.    

Response: The authors would like to thank you for the reviewer comment and new references have been updated in lines 94, 120 -122.

Reviewer 3 Report

Concerning our main remarks, corrections and suggestions about this work are:

1/ What is the novelty of this study and how does it differ from other studies previously published in the literature?

2/ Line 125-128 The most popular chemical degradation methods are photochemical and electrical chemical degradation, but there are also other types of  energy input to perform the degradation such as sonochemical degradation, microwave enhanced degradation, or other chemical-catalyzed degradation. with novel potential applications.. Should be supported by more refs. I suggest read https://doi.org/10.1016/j.colsurfa.2023.131509 and https://doi.org/10.1016/j.jwpe.2023.103925. This inclusion would further enhance the significance and context of your study.

3) What are the environmental sources and pathways through which PFOA contaminates different ecosystems?

4) How does PFOA bioaccumulate in aquatic life and humans, and what are its potential health effects?

5) What conclusions can be drawn from comparing the various chemical degradation methods discussed in terms of their efficiency, reaction conditions, and potential drawbacks?

6) How does the presence of short-chain PFCA as degradation products provide insights into the degree of defluorination during both chemical and biological degradation?

7) What are the three main types of degradation products that result from chemical degradation of PFOA?

Author Response

Response to Reviewers

We thank all the reviewers for their valuable comments and suggestions. All the comments have been addressed, and the manuscript has been revised suitably. The changes are highlighted in yellow in the revised manuscript.

Response to Reviewer 3

Concerning our main remarks, corrections and suggestions about this work are:

Comment#1. What is the novelty of this study and how does it differ from other studies previously published in the literature?

Response: The authors would like to thank you for the reviewer’s comment and the novelty has been included in the revised manuscript in research gap and innovation section (added in line 115-126).

Comment#2. Line 125-128 The most popular chemical degradation methods are photochemical and electrical chemical degradation, but there are also other types of  energy input to perform the degradation such as sonochemical degradation, microwave enhanced degradation, or other chemical-catalyzed degradation. With novel potential applications.. Should be supported by more refs. I suggest read https://doi.org/10.1016/j.colsurfa.2023.131509 and https://doi.org/10.1016/j.jwpe.2023.103925. This inclusion would further enhance the significance and context of your study

Response: The authors would like to thank you for the reviewer’s suggestion, however, the mentioned degradation methods were not reported for PFOA degradation in the literature. Further, the mentioned reference has been included in the revised manuscript as the reference is relevant to the current research.  (Added in lines 132-136).

Comment#3. What are the environmental sources and pathways through which PFOA contaminates different ecosystems?

Response: As per the reviewer’s comment, this information has been included in the revised manuscript (Updated in lines 65-69).

Comment#4. How does PFOA bioaccumulate in aquatic life and humans, and what are its potential health effects?

Response: As per the reviewer’s comment, this information has been included in the revised manuscript (Added in lines 77-78).

Comment#5. What conclusions can be drawn from comparing the various chemical degradation methods discussed in terms of their efficiency, reaction conditions, and potential drawbacks?

Response: As per the reviewer’s comment, this information has been included in the revised manuscript (This part of the information is available in the future outlook section lines 505-513)

Comment#6. How does the presence of short-chain PFCA as degradation products provide insights into the degree of defluorination during both chemical and biological degradation? Response: As per the reviewer’s comment, this information has been included in the revised manuscript (Updated in line 382-392).

Comment#7. What are the three main types of degradation products that result from chemical degradation of PFOA?

Response: As per the reviewer’s comment, this information has been included in the revised manuscript (Added in line 343, 344, it will be short chain PFCA, F and CO2).

Reviewer 4 Report

General Comments:

 The paper titled “Comparison between chemical and biological degradation processes for perfluorooctanoic acid” is a timely trending research focus due to the environmental persistence of the chemical and potential human health and toxicity concerns. I find this manuscript interesting to the readership of the journal.

I suggest expanding the introduction section discussing why it is important to focus on PFOA. Details of the physical-chemical properties of PFOAs, critical concerns, environmental fates, existing regulations for drinking water and wastewater effluents, and health concerns are needed to emphasize the importance of the manuscript. According to the authors, the novelty of the paper is the discussion of the biological degradation of PFOAs. But proportionally, discussion on biological degradation is limited. I suggest adding more discussion on specific biological treatment processes, if any.

The first part of the manuscript appears to be less organized compared to the latter part. There were several incomplete and unnecessarily lengthy sentences. Suggest improving the language to make the content clearer and concise.

Specific comments 

Line Number

Comment

15

Incomplete sentence. “and.”??

29

Plural is more appropriate “compounds”

30

Plural is more appropriate “compounds have been”

58

Irrelevant and informal language “but due to limitation of paragraph”. Suggest rephrasing.

66

Did you mean “invertebrates” ?

65-66

Specify which toxicity is this. E.g. acute or chronic, IC50, etc.

101

Incomplete sentence

96-119

This paragraph does not belong in the physical removal process section as it discusses chemical and biological degradation.

105-108

If the authors explicitly say the number of literatures, they should be cited here.  

113-115

Unclear sentence. vague

116-119

Novelty not discussed sufficiently

Introduction

Introduction could be expanded including physical-chemical properties of PFOAs. Better to discuss the importance, specific concerns, and environmental fate of PFOA in a separate paragraph

146

Should be “dropped to”

159-160

Under aerobic or anaerobic?

333

Repetitive content.

351

Rather than “reaction”, suggest “previous studies”

400

Did you mean “stable”?

402

“prof”?

481

Suggest discussing advantages and disadvantages of chemical and biological degradation in detail and major concerns in each pathway

In-text citation

Citations are inconsistent. Multiple citations are not merged.

Reference list

If placed in alphabetical order, no need to number them in the bibliography.

Writing style needs to be improved. There were several incomplete and unnecessarily lengthy sentences making it difficult to understand. Suggest improving the language to make the content clearer and concise. Suggest using past tense for discussing previous literature.  

Author Response

Response to Reviewers

We thank all the reviewers for their valuable comments and suggestions. All the comments have been addressed, and the manuscript has been revised suitably. The changes are highlighted in yellow in the revised manuscript.

Response to Reviewer 4

General Comments:

 The paper titled “Comparison between chemical and biological degradation processes for perfluorooctanoic acid” is a timely trending research focus due to the environmental persistence of the chemical and potential human health and toxicity concerns. I find this manuscript interesting to the readership of the journal.

Comment#1. suggest expanding the introduction section discussing why it is important to focus on PFOA. Details of the physical-chemical properties of PFOAs, critical concerns, environmental fates, existing regulations for drinking water and wastewater effluents, and health concerns are needed to emphasize the importance of the manuscript.

Response: The authors would like to thank you for the reviewer’s comment and the suggested information has been included in the revised manuscript (Updated in line 59-69)

Comment#2. According to the authors, the novelty of the paper is the discussion of the biological degradation of PFOAs. But proportionally, discussion on biological degradation is limited. I suggest adding more discussion on specific biological treatment processes, if any.

Response: The authors agree with the reviewer’s comment, however, the information and the research on biological degradation are limited for PFOA and the manuscript has included all the literature research provided on PFOA biological degradation (It is because the resources for the biological degradation is limited)

Comment#3. The first part of the manuscript appears to be less organized compared to the latter part. There were several incomplete and unnecessarily lengthy sentences. Suggest improving the language to make the content clearer and concise.

Response: Authors welcome this comment form the reviewer and all the necessary redundant statements were removed or edited suitably to improve the readability experience. (Updated in Lines 64-70)

Comment#4. Specific comments (At least provide a column and mentioned addressed at the designated place).

Line Number

Comment

15

Incomplete sentence. “and.”?? (Updated in line 16)

29

Plural is more appropriate “compounds” (Updated in line 30)

30

Plural is more appropriate “compounds have been” (Updated in line 31)

58

Irrelevant and informal language “but due to limitation of paragraph”. Suggest rephrasing. (Updated in line 59-60)

66

Did you mean “invertebrates” ? (Updated in line 70)

65-66

Specify which toxicity is this. E.g. acute or chronic, IC50, etc. (Updated in line 77-79)

101

Incomplete sentence (This sentence has been completed in line 106)

96-119

This paragraph does not belong in the physical removal process section as it discusses chemical and biological degradation. (This section has been updated in line 90)

105-108

If the authors explicitly say the number of literatures, they should be cited here.  (The number has been removed in line 118-120)

113-115

Unclear sentence. Vague (Updated in line 124-128)

116-119

Novelty not discussed sufficiently (Novelty has been discussed in line 124-128)

Introduction

Introduction could be expanded including physical-chemical properties of PFOAs. Better to discuss the importance, specific concerns, and environmental fate of PFOA in a separate paragraph (Updated in line 59 – 79)

146

Should be “dropped to” (Updated in line 151)

159-160

Under aerobic or anaerobic? (In both conditions)

333

Repetitive content. (Updated in line 345)

351

Rather than “reaction”, suggest “previous studies” (Updated in line 361)

400

Did you mean “stable”? (Since it is a stepwise reaction, “stage” is ok)

402

“prof”? (Updated in line 434)

481

Suggest discussing advantages and disadvantages of chemical and biological degradation in detail and major concerns in each pathway (It is in the future outlook section)

In-text citation

Citations are inconsistent. Multiple citations are not merged. (All the multiple citations have been merged)

Reference list

If placed in alphabetical order, no need to number them in the bibliography. (The reference list has been addressed according to the comment)

Comment#5 Comments on the Quality of English Language: Writing style needs to be improved. There were several incomplete and unnecessarily lengthy sentences making it difficult to understand. Suggest improving the language to make the content clearer and concise. Suggest using past tense for discussing previous literature.  

Response: The authors would like to thank you for the reviewer’s comment and the manuscript has been revised carefully.

Shine, please add lien number for each comment (Added in Lin…)

Round 2

Reviewer 1 Report

The authors have revised the manuscript.

Related with Comment 5, in the previous review:

Comment#5 The number of cited references can be enhanced for an effective review paper. Literature survey can be improved related with discussion on the past studies on the topic, the methods, the importance and the applications.  It would be better to add some additional related papers on water treatment techniques, with a few recommendations that would be helpful for readers' understanding the topic:

Research Progress on Process-Intensified Water Treatment Applications, Separations.  2022; 9(11), 353. https://doi.org/10.3390/separations9110353.

Response: The authors would like to thank you for the reviewer comment and new references have been updated in line 94, 120 -122.

The authors report that new references have been added. I recommend to check again, as the recommended reference has been forgotten I think.

Author Response

Response to Reviewers’ comments

The authors would like to thank the reviewers for their insightful comments and suggestions to improve the quality of the presented manuscript. All the changes have been included in the revised manuscript and are highlighted in yellow. In addition, the response to individual comments is presented as follows:

Response to Reviewer 1

Related with Comment 5, in the previous review:

Comment#5 The number of cited references can be enhanced for an effective review paper. Literature survey can be improved related with discussion on the past studies on the topic, the methods, the importance and the applications.  It would be better to add some additional related papers on water treatment techniques, with a few recommendations that would be helpful for readers' understanding the topic:

Research Progress on Process-Intensified Water Treatment Applications, Separations.  2022; 9(11), 353. https://doi.org/10.3390/separations9110353.

Response: The authors would like to thank you for the reviewer comment and new references have been updated in line 97, 123 -125.

Comment# The authors report that new references have been added. I recommend to check again, as the recommended reference has been forgotten I think.

Response: A recommended citation, labeled as "Speth, 2020," has been included at line 97, with the corresponding reference found in lines 752-755. Another citation, related to PFAS in wastewater treatment, has been revised to "Lenka 2021" at line 97, and the associated reference list is located in lines 691-693. The reference for the suggested review article, "Wackett, et al., 2021," has also been updated at line 125, and its corresponding citation can be found at lines 776-777. Additionally, at line 125, a new reference, "Zhang et al., 2022," has been introduced to discuss a previous study on PFAS, with its corresponding reference located in lines 803-804.

Reviewer 2 Report

Accept

Accept

Author Response

Response to Reviewer 2

Comment# Minor editing of English language required

Response: The authors would like to thank you for the reviewer’s comment and the manuscript has been revised carefully. The changes are highlighted in yellow in the revised manuscript.

Reviewer 3 Report

No comment

Author Response

Response to Reviewer 3

Comment# Editing of English language required

Response: The authors would like to thank you for the reviewer’s comment and the manuscript has been revised carefully. The changes are highlighted in yellow in the revised manuscript.

Reviewer 4 Report

Authors have adequately addressed majority of the concerns on the original version. I recommend the revised version for consideration for publication in this journal. Few minor corrections are listed below. 

Line Number

Comment

63

“it” should be “Its”

170

“use” should be “used”

339

Should be “want the bacteria to consume”

404

Did you mean “Laboratory”?

432

“Table”??

I noticed few grammartical and spelling mistakes in the manuscript. Suggest carefully reviewing the entire manuscript for such language errors/corrections. 

Author Response

Response to Reviewer 4

The authors have adequately addressed majority of the concerns on the original version. I recommend the revised version for consideration for publication in this journal. Few minor corrections are listed below. 

Line Number

Comment

63

“it” should be “Its” (The comment has been addressed in line 67)

170

“use” should be “used” (The comment has been addressed in line177)

339

Should be “want the bacteria to consume” (The comment has been addressed in line 369)

404

Did you mean “Laboratory”? (Yes, the word has been addressed as laboratory in line 444)

432

“Table”?? (The comment has been addressed as “stable” in line 469)

Comment# I noticed a few grammatical and spelling mistakes in the manuscript. Suggest carefully reviewing the entire manuscript for such language errors/corrections. 

Response: The authors would like to thank you for the reviewer’s comment and the manuscript has been revised carefully.